# Evidence of ghost plagioclase signature induced by kinetic fractionation of europium in the Earth's mantle

Romain Tilhac [1] ✉, Károly Hidas[2], Beñat Oliveira[3,4] & Carlos J. Garrido [1] ✉

Crustal recycling in the Earth's mantle is fingerprinted by trace-element and isotopic proxies in oceanic basalts. Positive Eu and Sr anomalies in primitive lavas and melt inclusions that are not otherwise enriched in $Al_2O_3$ are often interpreted as reflecting the presence of recycled, plagioclase-rich oceanic crust in their mantle source – referred to as "ghost plagioclase" signatures. Here, we report natural evidence of Eu anomalies and extreme crystal-scale heterogeneity developed kinetically in mantle peridotite clinopyroxene. Numerical modelling shows that diffusional fractionation between clinopyroxene and melts can account for this intra-crystal heterogeneity and generate Eu anomalies without requiring plagioclase. We demonstrate that kinetically induced Eu anomalies are likely to develop at temperatures, redox conditions and transport timescales compatible with the genesis of mid-ocean ridge and ocean island basalts. Our results show that, in the absence of converging lines of evidence such as radiogenic isotope data, ghost plagioclase signatures are not an unequivocal proxy for the presence of recycled crust in oceanic basalt sources.

Mid-ocean ridge (MORB) and ocean island (OIB) basalts are unique witnesses of the dynamic evolution of the Earth's mantle[1]. Trace elements in MORB and OIB lavas[2–5] and melt inclusions[6–8] as well as isotope tracers e.g. ref. [9] have shown that the source of oceanic basalts is a heterogeneous mixture of mantle peridotites and recycled crust. Owing to plagioclase-rich gabbroic cumulates e.g. ref. [10], the oceanic lower crust is characterised by positive europium (Eu) and strontium (Sr) anomalies [defined respectively as $(Eu/Eu^*)_N$ and $(Sr/Sr^*)_N > 1$ where $Eu^* = (Sm \cdot Gd)^{1/2}$, $Sr^* = (Ce \cdot Nd)^{1/2}$ and N denotes the normalisation to chondritic meteorites taken as a reference]. The presence of Eu and Sr anomalies in OIB melt inclusions and glasses from Hawaii[7,8,11,12] and Iceland[13,14] have been interpreted as reflecting the presence of recycled oceanic gabbro in the mantle source of these oceanic basalts. The term "ghost plagioclase" was first coined to refer to such interpretations based on positive Sr anomalies in melt inclusions from Mauna Loa olivines in the absence of $Al_2O_3$ enrichment[8]. Alternatively, Sr- and Pb-isotope compositions suggest that the ghost plagioclase signature could be explained by interaction with plagioclase-rich cumulates during magma ascent through modern oceanic crust[15,16]. Recycling of lower continental crust enriched in Eu was also invoked to account for the global mean value of $(Eu/Eu^*)_N = 1.03$ in primitive (i.e. MgO > 10 wt%) OIB glasses and the relative Eu depletion of the bulk continental crust[17]. In contrast, Niu and O'Hara[18] estimated from primitive MORB glasses that the missing Eu was retained in the depleted MORB mantle (DMM) during the extraction of the continental crust in a two-stage process involving partial melting of recycled oceanic crust.

[1]Instituto Andaluz de Ciencias de la Tierra (IACT), Consejo Superior de Investigaciones Científicas (CSIC) – Universidad de Granada (UGR), 18100 Armilla, Granada, Spain. [2]Departamento de Geología y Subsuelo, Centro Nacional Instituto Geológico y Minero de España (IGME), CSIC, Project Office of Granada, 18006 Granada, Spain. [3]ARC Centre of Excellence for Core to Crust Fluid Systems (CCFS) and GEMOC, Macquarie University, Sydney, NSW 2109, Australia. [4]Digital Health Solutions, Western Sydney Local Health District, North Parramatta NSW 2151, Australia. ✉ e-mail: romain.tilhac@csic.es; carlos.garrido@csic.es

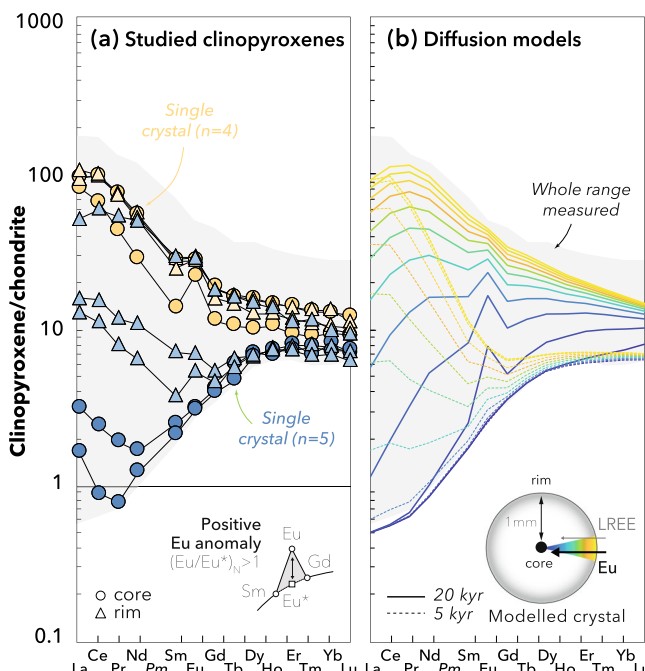

**Fig. 1 | Europium anomalies and progressive LREE enrichment in measured and modelled clinopyroxene crystals.** Measured core and rim compositions from single crystals and the whole measured range (**a**) are compared to core-to-rim profiles simulated after 5 kyr and 20 kyr (**b**) by the percolation-diffusion model (see Methods). Data are normalised to the chondrite compositions of McDonough and Sun[76]. All the analytical data are reported in Supplementary Data 1. *Pm* concentrations are interpolated.

The use of trace-element and isotopic proxies to assess the mantle source of primitive oceanic basalts assumes that melts are produced in chemical equilibrium with residual mantle minerals[19], but the universality of this assumption is debated[20–24]. The achievement of equilibrium conditions depends on the relative efficiency of melt segregation and transport with respect to the kinetics of equilibration processes, which is controlled by solid-state diffusion in the case of trace elements[25]. The experimentally determined diffusivities of rare-earth elements (REE) in mantle minerals are in the range of $10^{-23}$ to $10^{-18}$ m²·s⁻¹ [26–29]. In contrast, U-series disequilibria measured in young basalts require mantle upwelling velocity on the order of $10^{-10}$ to $10^{-9}$ m·s⁻¹ e.g. ref. [30] and melt flow velocity potentially exceeding $10^{-7}$ or $10^{-6}$ m·s⁻¹ [24,31–33]. These constraints indicate that chemical equilibrium may be the exception rather than the rule and that mantle melting and melt transport are most likely associated with diffusive fractionation of trace elements[23,34–37].

In addition to crustal recycling, positive anomalies of fast-diffusing cations such as Eu²⁺ and Sr²⁺ in oceanic basalts could be the result of disequilibrium processes. Recent data on primitive MORB glasses indeed yield an only slightly positive mean Eu anomaly [(Eu/Eu*)$_N$ = 1.02[38]] consistent with most of the previous global MORB estimates[39–41], except that of Niu and O'Hara[18]. Tang, McDonough[38] argued that such anomalies can be accounted for by disequilibrium melting whereby Eu preferentially enters the melt to produce positive anomalies, leaving negative Eu anomalies and zoning towards light REE (LREE)-depleted rims in residual clinopyroxenes[35]. Despite increasing evidence of disequilibrium processes in the Earth's mantle, large crystal-scale REE heterogeneities diagnostic of disequilibrium melting processes have not been documented so far in mantle peridotites. Only limited zoning towards LREE-enriched rims was reported in clinopyroxene from abyssal peridotites[42].

Here we provide compelling natural evidence from clinopyroxene in high-temperature mantle peridotites of extreme, intracrystalline REE variability and transient Eu anomalies produced by kinetic fractionation. Numerical modelling indicates that diffusional disequilibrium can account for the observed variability, indicating that ghost plagioclase signatures may develop in the source of primitive oceanic basalts without requiring the involvement of plagioclase-rich lithologies.

## Results and discussion

### Extreme crystal-scale heterogeneity in mantle peridotite clinopyroxene

The studied samples are fresh spinel-facies mantle xenoliths rapidly brought to the surface by the Plio-Quaternary alkali basalts of the Oran volcanic field in the Tell Atlas, NW Algeria[43]. These xenoliths preserve high equilibration temperatures (i.e. up to 1165 °C) and are free of metasomatic mineral assemblages (sample descriptions are provided in Supplementary Information). They mostly range from lherzolites to harzburgites, which are thought to be dominant lithologies in the mantle source of oceanic basalts. In-situ analyses by laser-ablation inductively coupled plasma mass spectrometry (LA-ICP-MS; see Methods) reveal that the clinopyroxenes exhibit a wide range of REE concentrations over nearly four orders of magnitude (Fig. 1). They show flat but variable heavy REE (HREE) distributions, overlapping those in clinopyroxenes from abyssal and ophiolitic peridotites, and highly variable LREE enrichment comparable to, or even exceeding, that observed in most mantle xenoliths and orogenic peridotites[44]. Preferential enrichment of the LREE is commonly ascribed to chromatographic fractionation during porous-flow melt percolation upon which concentration fronts migrate at a rate proportional to the element's ability (increasing from HREE to LREE) to partition into melt[45]. However, the studied clinopyroxenes also exhibit a strong intercrystal variability within individual samples and extreme core-to-rim zoning at the mm-scale (Fig. 1a), which constitutes among the largest trace-element variations at this length scale ever observed in mantle peridotite clinopyroxene e.g. ref. [42]. Such compositional gradients cannot be produced by chromatography because this process requires length scales of several metres to hundreds of metres. To the contrary, the extreme crystal-scale heterogeneities observed point to a kinetic process operating within the crystals as the driving mechanism.

A clue to understanding the origin of the extreme heterogeneity observed in the studied clinopyroxenes is that core-to-rim LREE fractionation is accompanied by variable Eu anomalies with (Eu/Eu*)$_N$ ranging from 0.8 to 1.8 (Fig. 2). On the one hand, subsolidus and melt-present re-equilibration of clinopyroxene in the presence of plagioclase is excluded because it would result in (Eu/Eu*)$_N$ < 1 e.g. refs. [42,46], which is not observed in our samples. On the other hand, if the clinopyroxene had equilibrated with a melt with a positive Eu anomaly, for instance derived from a former plagioclase-bearing protolith e.g. ref. [47], a positive correlation would be observed between (Eu/Eu*)$_N$ and LREE enrichment towards the clinopyroxene rims. However, while (Eu/Eu*)$_N$ does increase with LREE enrichment at low Eu contents, it markedly decreases with further enrichment at higher Eu contents (Fig. 2). Furthermore, the greatest Eu anomalies are not observed in the clinopyroxene rims and neither correspond to the strongest LREE enrichment or Eu contents. This systematic evolution of the (Eu/Eu*)$_N$ ratio excludes that Eu anomalies in our samples reflect equilibrium processes in the presence of plagioclase or melt derived from plagioclase-bearing lithologies. Instead, these Eu anomalies were formed as a response to strong compositional gradients between LREE-depleted clinopyroxene and LREE-enriched melt both devoid of Eu anomalies.

### Development of Eu anomalies by kinetic fractionation

The main limiting factor controlling the extent of mineral-melt trace-element re-equilibration is solid-state diffusion in crystals[25]. To first order, the diffusivities of REE decrease with the ionic radius – ranging

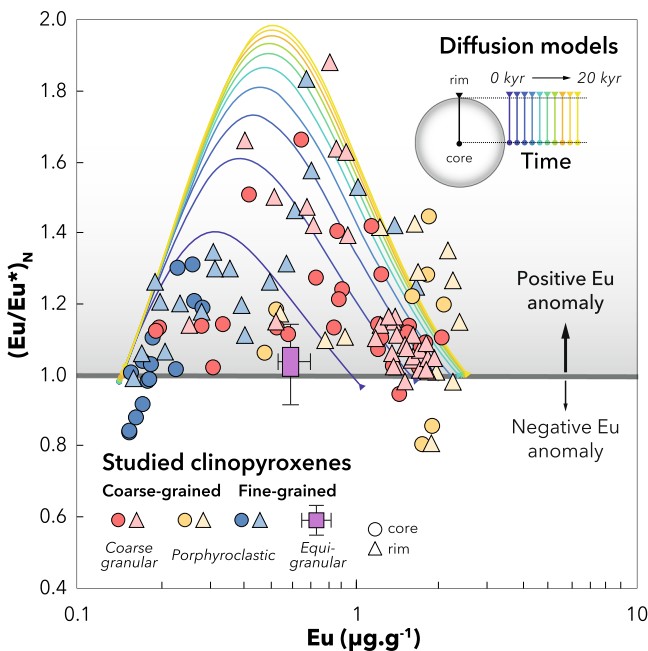

**Fig. 2 | Transient evolution of the Eu anomalies in clinopyroxene.** Data points are colour-coded based on the textural classification used in Fig. 4 and compared to the core-to-rim profile simulated within a single clinopyroxene crystal after different times (0–20 kyr) by the percolation-diffusion model (see Methods). For the equigranular population, the mean value and external standard deviation are shown.

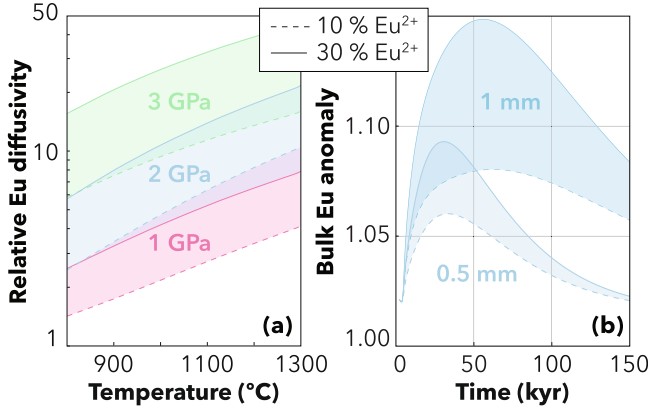

**Fig. 3 | Relative Eu diffusivity and development timescale of kinetic Eu anomalies.** The relative diffusivity (**a**) is calculated as the diffusivity of a mixture ($Eu^{mix}$) of divalent and trivalent Eu divided by the diffusivity of pure $Eu^{3+}$. The development timescale (**b**) is shown as the time evolution of the bulk anomaly in clinopyroxene (calculated as the integral of the core-to-rim profile in spherical coordinates) modelled at 2 GPa and 1300 °C.

from 110–117 Å for $LREE^{3+}$ to 100–108 Å in $HREE^{3+}$ – and increase with temperature. In clinopyroxene, the experimentally determined diffusion of the LREE lanthanum (La) is ~35 times slower than that of the HREE ytterbium (Yb)[27]. In contrast to other REE which mostly form trivalent cations, Eu may occur in two oxidation states ($Eu^{3+}$ and $Eu^{2+}$). Owing to the presence of divalent Eu ($Eu^{2+}$), Eu diffuses over an order of magnitude faster under reducing than under oxidising conditions[28]. Although the diffusivity of pure $Eu^{2+}$ has not yet been experimentally measured, it can be approximated by the diffusivity of $Sr^{2+}$ [38], which is orders of magnitude faster than that of $REE^{3+}$ even using the most conservative estimates[48]. Consequently, even small fractions of $Eu^{2+}$ can substantially enhance the bulk diffusivity of Eu in clinopyroxene and this effect increases with pressure and temperature (Fig. 3a). At 1200 °C and 1.5 GPa, the presence of 10–30% of divalent Eu increases by a factor of ~5–12 the bulk diffusivity of Eu and this factor reaches ~8–22 at 1300 °C and 2.0 GPa, which are conditions encountered in the mantle source of MORB and OIB.

To test whether the positive Eu anomalies observed result from diffusive fractionation of REE, we numerically simulated the diffusional re-equilibration of LREE-depleted clinopyroxene percolated by a LREE-enriched melt, both devoid of Eu anomalies (see Methods). The results show that a strong transient zoning develops from core to rim in clinopyroxene (Fig. 1b). After 5 kyr, the model results reproduce the positive Eu anomalies over most of the range of moderate LREE enrichment observed in clinopyroxenes. After 20 kyr, the modelled clinopyroxene preserves a depleted LREE pattern with no Eu anomaly in the core while a positive Eu anomaly develops and disappears as LREE enrichment increases towards the rim. This core-to-rim zoning reproduces strikingly well even the most heterogeneous clinopyroxenes in our samples (Fig. 1a). The temporal evolution of this zoning pattern also overlaps with the whole range of $(Eu/Eu^*)_N$ measured in the studied clinopyroxenes (Fig. 2), suggesting that different samples experienced variable extents of diffusive re-equilibration. Our percolation-diffusion model thus reproduces both the variability

observed between samples and the fractionation observed within single crystals.

The role of diffusion is also supported by a grain-size distribution control on the REE concentrations, as documented by textural observations in the studied samples[43]. Peridotites with coarse-granular and equigranular textures exhibit the least LREE-enriched patterns and LREE enrichment increases from coarse- to fine-grained samples (Fig. 4a). In contrast, peridotites with porphyroclastic textures exhibit highly variable LREE enrichment, increasing from fine- to coarse-grained samples. Significant HREE variability is observed in coarse-grained porphyroclastic samples (Supplementary Fig. S1), which is absent from fine-grained samples because the fast-diffusing HREE are buffered by the fine-grained population (i.e. with large surface-to-volume ratio). Importantly, this textural control is also striking in terms of Eu anomalies. Variably positive $(Eu/Eu^*)_N$ are found in porphyroclastic [$(Eu/Eu^*)_N = 1.2 \pm 0.6$], and to a lesser extent, coarse-granular samples, but no anomaly is observed in equigranular samples [$(Eu/Eu^*)_N = 1.0 \pm 0.1$] which re-equilibrate more readily (Fig. 4b).

## Implications for the source of oceanic basalts

Our observations indicate that diffusive fractionation of REE can effectively produce positive Eu anomalies in high-temperature peridotite clinopyroxene. Suitable conditions for this kinetic process to take place in the Earth's mantle require the presence of a sufficient proportion of $Eu^{2+}/Eu^{tot}$ and transient interaction with highly LREE-enriched melt. The proportion of $Eu^{2+}/Eu^{tot}$ depends on the redox state of the system considered and thus on oxygen fugacity ($fO_2$), as discussed by Tang, McDonough[38]. Depending on the estimates, MORB sources are within one log-$fO_2$ unit of the fayalite-magnetite-quartz (FMQ) redox buffer[49–51] and have $Eu^{2+}/Eu^{tot}$ ranging between ~5 and 30%[51–53]. Redox conditions are comparable or slightly more oxidising in OIB sources[54,55]. These estimates are in good agreement with $Fe^{3+}/Fe^{2+}$ measurements in spinel peridotites suggesting that the upper mantle in the ~30–60-km depth range is within 2 log-$fO_2$ units of FMQ. At greater depths, because $Fe^{3+}/Fe^{2+}$ is pressure-dependent, $fO_2$ most likely decreases further as recorded in garnet peridotites plotting 1–4 log units below FMQ[54]. Overall, both MORB and OIB sources, which are mainly located within the stability field of spinel and/or garnet peridotite, provide suitable redox conditions for significant $Eu^{2+}$ to be present and Eu to diffuse faster than other REE. This is particularly true in the deepest regions where the relative diffusivity of Eu is the highest (Fig. 3). As decompression melting is the dominant melt generation mechanism in oceanic basalts, these deep regions are also

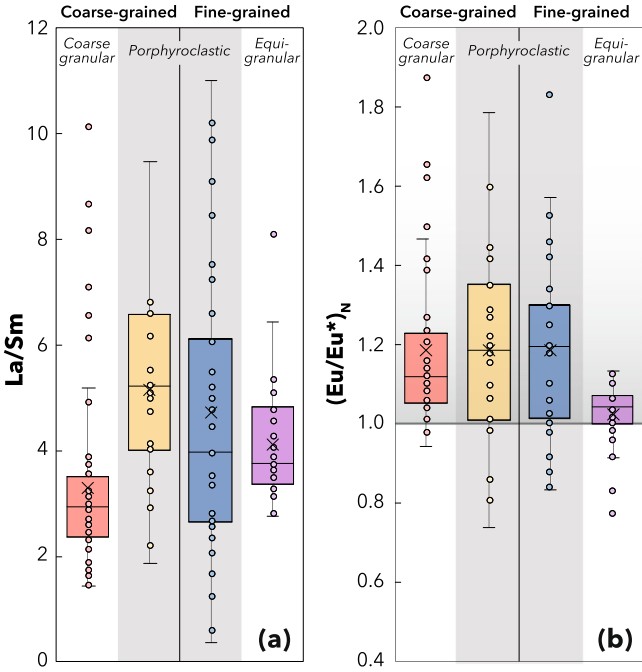

**Fig. 4 | Grain-size control on the development of Eu anomalies and LREE enrichment by diffusive fractionation.** Box-and-whisker plots showing that (**a**) LREE enrichment in clinopyroxene increases from coarse- to fine-grained samples and is most dispersed in porphyroclastic samples, and (**b**) Eu anomalies are consistently absent from fine-grained equigranular samples. The grain-size distributions for each textural group are reported in Supplementary Fig. S2. Whiskers correspond to 1.5 of the interquartile range; the mean value is shown by a cross.

where the presence of LREE-enriched melts is promoted by low degrees of melting. This mechanism is documented by the extreme compositional variability and LREE enrichment in near-axis lavas whose primary melts escape the melting region at greater depths than aggregated melts erupted on-axis[56] The deep mantle source of oceanic basalts thus provides preferential redox conditions and melt compositions for diffusive REE fractionation to produce positive Eu anomalies in peridotite clinopyroxene.

To assess the viability of this process, we numerically modelled the time evolution of bulk Eu anomalies produced by diffusive fractionation (Fig. 3b). We found that 1-mm crystals acquire a bulk (Eu/Eu*)$_N$ that exceeds 1.07 in 10–20 kyr depending on $Eu^{2+}/Eu^{tot}$ considered, and even 0.5-mm crystals are sufficient to produce anomalies comparable to the highest global MORB estimates[18] and the range observed in OIB[17]. These timescales are entirely compatible with available temporal constraints on the genesis of oceanic basalts. In mid-ocean ridges, significant diffusive fractionation probably occurs in the low-degree, hydrous melting region that lies beneath the dry peridotite solidus[57]. If we consider a reasonable range of water content of 0–200 µg·g$^{-1}$ in the ambient mantle[58], we can estimate this region to extend over a depth range of ~30 km, depending on the potential temperature considered[59]. Assuming upwelling rates of 1–10 cm·yr$^{-1}$, comparable to the spreading rates of medium to fast mid-ocean ridges[60], we calculate that mantle peridotites ascend through the hydrous melting region in >300 kyr, although the rheological effect of water probably implies faster upwelling beneath the dry peridotite solidus[61,62]. This duration is compatible with temporal constraints provided by U-series disequilibria e.g. ref. [30] and largely sufficient for diffusive fractionation to produce significant Eu anomalies in clinopyroxene (Fig. 3b). As for OIB, estimates of mantle upwelling rates based on U-series disequilibria in Iceland and Hawaii are very variable, ranging between 0.1 and 100 cm·yr$^{-1}$ [63,64]. In the fastest scenario, these upwelling rates provide a minimum timescale of ~100 kyr assuming a

depth of minimum depth of 100 km for the onset of melting[65]. Even using conservative estimates of solidus depths and ascent rates, these temporal constraints are compatible with the development timescales of kinetically induced positive Eu anomalies overlapping most of the MORB and OIB variability (Fig. 5a).

## The ghost plagioclase as a disequilibrium signature?

The development of positive Eu anomalies *via* diffusive fractionation in mantle peridotite clinopyroxene is likely to occur in MORB and OIB sources at depths where low degrees of melting promote the presence of LREE-enriched melts (Fig. 5b, in blue). This situation is the counterpart of the disequilibrium melting scenario hypothesised by Tang, McDonough[38], whereby positive anomalies are instead produced in the melt (Fig. 5b, in red). The disequilibrium melting scenario is expected to leave LREE-depleted residual clinopyroxenes with negative Eu anomalies, which have not been reported to date and might not easily be documented because such clinopyroxenes are prone to be overprinted. Rather than contradicting this scenario, our observations of transient positive Eu anomalies represent a unique snapshot providing natural evidence that disequilibrium processes can strongly fractionate fast-diffusing divalent cations in the upper mantle. Disequilibrium melting is merely one of two non-mutually exclusive scenarios (Fig. 5b), which may occur to various extents in basaltic sources and contribute to the variability of Eu anomalies observed in primitive lavas (Fig. 5a). For instance, in the melting regime of MORB, we envisage a two-stage process whereby positive anomalies first develop in clinopyroxene in the region of low degrees of hydrous melting that prevails at depth (Fig. 5c). As dry melting proceeds at shallower depths, these anomalies can be transferred to the melt and potentially be enhanced by disequilibrium melting. We thus concur with Tang, McDonough[38] that Eu and Sr anomalies in MORB do not imply the enrichment of the DMM in Eu and Sr and rather reflect the role of disequilibrium processes. Interestingly, relative depletion in the fast-diffusing HREE and $^{230}Th/^{238}U$ disequilibrium in MORB, traditionally interpreted as a garnet signature[2], are probably also due to diffusive fractionation[23,36].

In OIB glasses and melt inclusions, positive Eu and Sr anomalies are not sufficient to identify recycled plagioclase-bearing lithologies in the absence of converging lines of evidence such as elevated $Al_2O_3$ or enriched Pb (or Sr) isotopes[15]. In such cases, the ghost plagioclase signature most likely reflects the interplay of disequilibrium processes during partial melting and melt transport, in addition to other kinetic processes taking place during crustal contamination and the post-entrapment evolution of melt inclusions[66,67]. Nonetheless, the presence of lithologies derived from oceanic recycling, such as eclogites and garnet pyroxenites[68,69], may indirectly promote the kinetic development of Eu or Sr anomalies. These lithologies can indeed produce highly reactive melts that are, despite their poor extractability, in strong disequilibrium with the peridotite minerals[70]. Such strong chemical gradients are prone to promote diffusive fractionation e.g. ref. [4], which is likely to contribute to the development of anomalies that could be falsely interpreted as reflecting the involvement of plagioclase.

## Methods

### In-situ REE analyses in clinopyroxene

The REE compositions of clinopyroxene cores and rims were obtained in situ by LA-ICP-MS at the Instituto Andaluz de Ciencias de la Tierra (IACT, Granada, Spain) using an Agilent 8800 QQQ ICP-MS attached to a CETAC-Photon Machines Analyte G2 system equipped with a 193-nm Excimer laser. The laser was fired using a circular beam size of 85-µm diameter at a pulse rate of 10 Hz and an energy density of 9.28 J·cm$^{-2}$. Each analytical sequence included 30 s of sample collection and 20 s of background. NIST SRM610 was used as an external standard and $^{43}Ca$ was taken as the internal standard. The data-reduction software Iolite

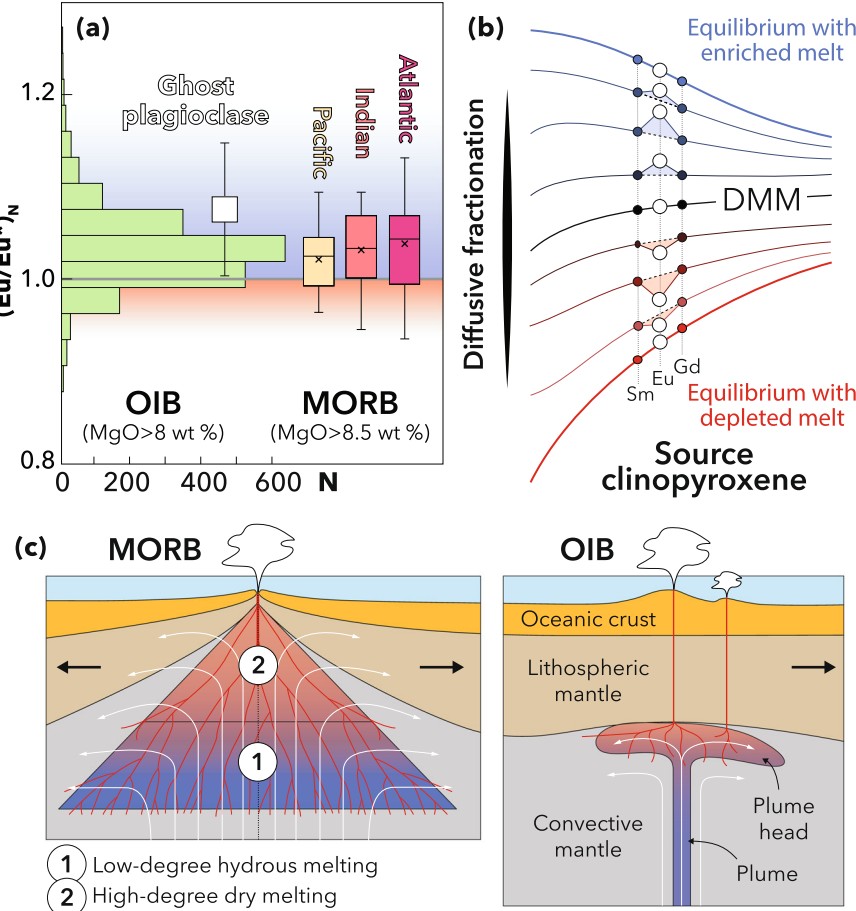

**Fig. 5 | Development of kinetic Eu anomalies in the mantle source of MORB and OIB. a** Global distribution of Eu anomalies in MORB and OIB glasses compared to Hawaiian melt inclusions with a ghost plagioclase signature. **b** Conceptual representation of the development of positive and negative Eu anomalies by diffusive fractionation in bulk clinopyroxene depending on the melt compositions. Positive Eu anomalies can be kinetically produced in clinopyroxene upon interaction with enriched melts (in blue), as documented by the Oran clinopyroxene data. Disequilibrium melting can also produce (or enhance) positive Eu anomalies in the melt, as envisaged by Tang et al.[38], leaving negative Eu anomalies in clinopyroxene (in red). **c** Regions with suitable conditions for the presence of enriched (low-degree) and depleted (high-degree) melts in the mantle source of MORB and OIB. Data for MORB are from ref. [38]; whiskers correspond to 1.5 of the interquartile range; the mean value is shown by a cross. Data for OIB are from the GeoRoc database (http://georoc.mpch-mainz.gwdg.de/georoc/) filtered for MgO >8 wt%. Data for melt inclusions are from Mauna Loa olivines[8] (±1σ). DMM, Depleted MORB Mantle.

was used to calibrate the fractionation induced by ablation, transportation and excitation processes and the matrix effect[71]. Analyses of USGS standards BCR-2G and BIR-1G were included as unknown samples in each batch for quality control and yielded compositions within uncertainty of the GeoRem recommended values (http://georem.mpch-mainz.gwdg.de).

**Percolation-diffusion modelling**

A numerical model was used to simulate the diffusion-controlled exchange of trace elements during the percolation a melt through a solid matrix following Tilhac, Oliveira[72]. The modelled system is a 1-D domain composed of solid mineral grains ($s$), approximated by spheres, and a percolating melt ($f$). The melt is assumed to instantaneously reach equilibrium with the mineral surface, which induces compositional gradients within the grains. This disequilibrium then tends to re-equilibrate with time by solid-state diffusion. The concentration of a trace element $e$ within a spherical grain of radius $R_s$ is given by Fick's second law (in spherical coordinates):

$$\frac{Dc_s^e}{Dt} = D_s^e \left( \frac{\partial^2 c_s^e}{\partial r^2} + \frac{2}{r} \frac{\partial c_s^e}{\partial r} \right) \tag{1}$$

where $c_s^e = c_s^e(r_s, t)$ is the concentration at a distance $r_s$ from the centre of the grain and $D_s^e$ is the diffusion coefficient of the trace element $e$. Initial and boundary conditions of Eq. (1) are based on the assumption of initial chemical equilibrium stated above (i.e. instantaneous equilibration between melt and grain surface). In addition, radial symmetry is imposed on the compositional zoning within grains. These conditions read as follows:

$$\frac{c_{s,0}^e}{c_{f,0}^e} = K_s^e c_s^e \big|_{r=R_s} = K_s^e c_f^e \frac{\partial c_s^e}{\partial r} \big|_{r=0} = 0 \tag{2}$$

where $K_s^e$ is the partition coefficient and $c_f^e$ is the concentration of the trace element $e$ in the melt. The mass-balance equation for element $e$ in the melt is:

$$\frac{\partial \phi \rho_f c_f^e}{\partial t} + \nabla \cdot \left( \phi \rho_f c_f^e v_f \right) = \Gamma^e \tag{3}$$

where $\phi$ is the porosity (i.e. melt proportion in volume), $\rho$ is the density, $v$ is the velocity, $\Gamma^e$ is the mass-transfer rate between solid grains and melt due to phase transformation (i.e. partial melting or crystallisation; in this case, none) and diffusional re-equilibration within solid grains. $\Gamma^e$ is computed by evaluating the gain/loss of a given trace

element $e$ within individual grains using the following integral:

$$\Gamma^e = -\frac{D_s}{Dt}\left[n_s\rho_s\int_0^{R_s(t)}4\pi r^2 c_s^e dr\right] \qquad (4)$$

where $n_s = \frac{(1-\phi)}{4/3\pi R_s^3}$ refers to the number of grains per unit of volume[34,73]. Equations (1)–(4) are solved numerically using a Eulerian–Lagrangian finite element method[74]. At each time step, the concentrations in the solid grains (Eqs. (1) and (2)) and in the melt (Eqs. (3)–(4)) are computed iteratively until convergence is reached. An irregular mesh is used for the diffusion problem (i.e. finer spatial discretization near the grain boundary).

To test whether the positive Eu anomalies measured in clinopyroxene result from diffusive fractionation of REE, we aimed to reproduce the range of Eu anomalies observed in the whole dataset as well as the most extreme gradient observed within single grains. The REE compositions were modelled in spherical clinopyroxene grains with a radius of 1 mm in the presence of finer-grained population (0.01 mm), comparable to the binary grain-size distribution (i.e. porphyroclastic texture; Supplementary Fig. S2) of the samples that exhibit the greatest variability of Eu anomaly (Fig. 4b). We used a constant melt porosity of 1% and a relative melt velocity of $1.6\cdot10^{-9}$ m·s$^{-1}$ following Tilhac, Oliveira[72]. The clinopyroxene starting composition, initially homogeneous, was taken as the most LREE-depleted core and the melt composition was calculated to be in equilibrium with the most enriched rim, which is highly LREE-enriched considering the strong incompatibility of LREE. The initial REE patterns of clinopyroxene and melt were smoothed to be both completely devoid of Eu anomaly. Partition coefficients were taken from Sun and Liang[75]; REE diffusivities ($D$) were calculated based on the Arrhenius relationship experimentally determined in clinopyroxene[27]:

$$D = D_0 e^{\left[\frac{-\varepsilon + PV}{RT}\right]} \qquad (5)$$

with $V = 10^{-5}$ m$^3$·mol, $P = 1.2$ GPa, $T = 1200$ °C and $R$ is the gas constant. $D_0$ and $\varepsilon$ are given in Supplementary Table S1. The diffusivity of Eu was calculated following Tang, McDonough[38] as a mixture of divalent and trivalent Eu using the diffusivity of Sr$^{2+}$ as a proxy for Eu$^{2+}$. The Arrhenius relationship experimentally determined by Sneeringer, Hart[48] in synthetic diopside crystals was used in order to get a conservative estimate of Sr$^{2+}$ diffusivity, whereas natural samples yield diffusivities that are two orders of magnitude faster. All the model parameters are summarised in Supplementary Table S1. Results shown in Fig. 1b were obtained after 5 and 20 kyr for the clinopyroxenes respectively located 30 m and ~0 m from the bottom of the 1-D domain, respectively, in order to show the competing effect of chromatographic and kinetic fractionation. Note that the model mostly yields positive Eu anomalies associated with moderate LREE depletions except during the early stages of percolation (e.g. at 5 kyr), whereas the Oran clinopyroxene data exhibit a range of positive Eu anomalies that may coexist in a single grain with moderate LREE enrichments. More extreme percolation-diffusion parameters would be needed to fully reproduce the range observed within such grains, but we preferred to use conservative experimental constraints on diffusivities and model settings generally extrapolatable to oceanic basalts source. Results shown in Fig. 2 are core-to-rim profiles colour-coded in time for a single coarse clinopyroxene located at the bottom of the 1-D domain after time increments of 0 to 20 kyr.

## Data availability

The authors declare that all data generated in this study are included in Supplementary Data 1. Source data are provided with this paper.

## Code availability

A MATLAB implementation of the numerical model described in the Methods is available on GitHub (https://github.com/romaintilhac/percolation-diffusion.git) and Zenodo (https://doi.org/10.5281/zenodo.8475).

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

## Acknowledgements

We are grateful to J.-L. Bodinier, J.-M. Dautria, A. Louni-Hacini and A. Azzouni-Sekkal for field work and sampling. This study was supported by the Consejo Superior de Investigaciones Científicas (CSIC) and by the Ministerio de Ciencia e Innovación and the Agencia Estatal de Investigación through grants IJC2020-044739 (to R.T.) and AEI-PID2021-122792NA-I00 (to R.T.) funded by MCIN/AEI/10.13039/501100011033, "ESF, Investing in your future" and "ERDF, A way of making Europe". This is contribution 1754 from the ARC Centre of Excellence for Core to Crust Fluid Systems (www.ccfs.mq.edu.au) and 1523 from the GEMOC Key Centre (www.gemoc.mq.edu.au).

## Author contributions

R.T. and C.J.G. co-designed this project. R.T. and B.O conducted the numerical modelling. K.H. collected the trace-element and textural data. R.T and C.J.G. wrote the manuscript with contributions from K.H. and B.O.

## Competing interests

The authors declare no competing interests.
