## [Peer review file · Nature Communications]

REVIEWER COMMENTS

Reviewer #1 (Remarks to the Author):

Review for Tilhac et al., Nature Communications

This paper by Tilhac and coauthors documents mineral scale evidence for chemical disequilibrium in the mantle sources of MORB and OIB. In particular, the presence of positive Eu anomalies only in clinopyroxene domains with mildly elevated LREE (Fig. 1a) and medium Eu concentrations (Fig. 2) is compelling. The numerical modeling done by the authors also reproduces the observations on a first order. The authors used their findings to argue against the suggestion of recycled plagioclase-rich lower crust (ghost plagioclase) in the mantle. Overall, I agree that positive Eu or Sr anomalies, in the absence of other evidence, cannot be taken as definitive evidence for recycled plagioclase. This is something we also suggested a few years ago, but we did not have the sort of intriguing evidence reported here. The paper is well written. I have a few concerns that need to be addressed.

First of all, how do we exclude the possibility that the disequilibrium Eu signatures were produced by the interactions between the xenoliths and the recent alkali basaltic magmas that carried the xenoliths to the surface? This is important. If the disequilibrium signatures were produced by these recent melt-mineral interactions, it would compromise the connection between the authors' observations and the disequilibrium mantle processes relevant to MORB and OIB genesis.

Second, in the measured clinopyroxene domains, strongly positive Eu anomalies coexist with moderate LREE enrichments (Fig. 1a) whereas in the modeling results strongly positive Eu anomalies are always associated with moderate LREE depletions (Fig. 1b). Why? Is it because the melt component assumed in the modeling is not sufficiently enriched in LREE? Or the timescales chosen (20 kyr and 5 kyr) did not capture the transient signatures?

Lines 52-53. I'm not sure Niu and O'Hara (2009) meant to suggest that the missing Eu was retained in the DMM during the extraction of the continental crust. They probably also interpreted it as evidence for some sort of crustal recycling instead of a single-stage process.

Fig. 3b. Would be more intuitive to plot bulk Eu anomaly as a function of time instead of the opposite?

Reviewer #2 (Remarks to the Author):

The presence of Eu positive anomalies, frequently coupled to prominent Sr (and Ba) enrichments, in primitive oceanic basalts, glasses and melt inclusions from both MORB and OIB settings has been commonly attributed to the former presence of plagioclase in gabbroic cumulates from the oceanic crust recycled back into the mantle (the so called “ghost plagioclase signature”). Other explanations claimed interaction with plagioclase-bearing cumulate sequences of the modern oceanic lithosphere or took into account disequilibrium processes during partial melting. The numerical model proposed in this paper, applied to a natural occurrence of peridotite xenoliths from the Oran volcanic field (Algeria), provides a brilliant alternative solution to generate such Eu anomalies, when the crustal recycling involvement is not adequately supported by other concomitant evidences (e.g. Pb or O isotopes).

In particular, the most relevant result achieved in this paper demonstrates that the Eu enrichment in clinopyroxenes from potential mantle sources of oceanic basalts may be reproduced by a diffusive fractionation process. The authors adopt the Oran xenoliths as a proxy of MORB-OIB mantle sources characterized by the preservation of transient positive Eu anomaly in clinopyroxene and reproduce through numerical simulation the striking intrasample variability of their REE patterns (or even zoning of single crystals).

In my opinion, this is an elegant, well written high quality paper based on a solid and sophisticated trace element modelling. I do not have particular remarks about manuscript organization and clarity of the abstract. I agree with the authors that the Eu anomalies in oceanic basalts may have different origins, not necessarily related to a role of plagioclase. The meaning of Eu enrichments in primitive melts (especially from MORB settings) may be reconsidered in the light of the new model presented here. Overall, the consistency of the adopted model, the originality of the approach and the accuracy of data treatment are remarkable and this work may certainly stimulate further debate on the paradigm of crustal recycling relying on trace element signatures like Eu and Sr.

The most relevant previous works have been adequately considered with few exceptions that I will underline below. To my knowledge, the conclusions of this manuscript are absolutely novel and original.

I am basically favourable to the publication of this manuscript provided that the authors address some issues reported in the following:

- Although no such systematic and detailed description of the Eu anomalies nor modelling have been attempted before, the occurrence of such anomalies in clinopyroxenes from spinel orogenic peridotites have been already reported elsewhere (e.g. see Rampone et al., 1995, *J.Petrology*, DOI: 10.1093/petrology/36.1.81; Nain et al., 2018, <https://doi.org/10.1016/j.lithos.2018.04.001>; Marchesi et

al. 2013, <http://dx.doi.org/10.1016/j.epsl.2012.11.047>), so I suggest to add a brief report in this regard. In the Ronda lherzolites, the geochemical signature of ghost plagioclase in the form of positive Eu and Sr anomalies in whole-rock and clinopyroxene was attributed to the transfer of a low-pressure crustal imprint from the adjacent recycled pyroxenite to hybridized peridotite, whereas in the other cases no straightforward explanations were provided.

-The analytical methodology is sound and the quality of data is very good. However, I suggest to add in Appendix A a table including the REE composition of the clinopyroxenes (average or representative values) used for the calculations. This would help to make the results reproducible. Also, a brief description of the samples taken from Hidas et al., 2019 and employed for the modelling could be useful and more informative for the reader.

-The procedures adopted for the calculations and data treatment is exhaustively explained in the Appendix B but the results are not easily reproducible. Several references are cited (Oliveira et al., 2016, 2018, 2020; Tihac et al., 2020) but no clear reference to a downloadable MATLAB code is reported. In alternative, a sentence like “A MATLAB (?) implementation of the described procedure is available from the authors upon request” could be added.

-The Eu positive anomalies of the melts are generally coupled to Sr (and Ba) enrichments (Hofmann and Jochum, 1996; Chauvel et al., 2000; Kokfelt et al., 2006; Peterson et al., 2014; Tang et al., 2017; Anderson et al., 2021). However, Sr abundances were not taken into account for modelling of diffusive fractionation. It would be interesting to see how Sr (which has been certainly measured in the cpx from the xenoliths) behaves during the kinetic fractionation affecting the REE.

-It looks like that the occurrence of peridotite sources with (transient) Eu enrichments in their cpx may explain the generation of melts with this signature. Since the size of the Eu anomaly seems correlated with LREE enrichment of the cpx, I would expect that the derivative Eu-enriched melts are not LREE-depleted like common N-MORB. If we consider a large amount of literature data, is there a correlation between Eu/Eu^* and LREE fractionation (e.g. LaN/SmN)?

-The authors convincingly explain that in the MORB (and OIB) sources timescale of melt generation and redox conditions are appropriate for their model and that the LREE-enriched melts required for the diffusive fractionation could form (in the case of MORB) in a hydrous melting region deeper than the dry melting region. However, the authors state (lines 240-241) that “significant diffusive fractionation probably occurs in the low-degree, hydrous melting region that lies beneath dry peridotite solidus (Asimow and Langmuir, 2003)”. Afterwards, the rocks affected by these “transient” Eu anomalies should be involved in the dry decompression melting (and, consequently, the LREE enrichment are lost). However, is this scenario is the modelling realistic? The diffusive fractionation model (Appendix B) assumes no concomitant melting, which seem in contrast with the sentence above. Otherwise, an open

system melting (e.g. Brunelli et al., 2014, GCA) including the effects of kinetic fractionation could be more appropriate?

-From Appendix B: to the purpose of the calculations “..it is assumed that no chemical reaction occurs between peridotite and melt”. If I understand properly, the cpx (or some cpx) from the xenoliths derive from opx replacement by reactive melt percolation, as supported by EBSD data (Hidas et al., 2019). Does this affect somehow the model?

-Lines 301-309: melting of silica-excess recycled mafic lithologies produces relatively SiO₂-rich melts which are highly reactive towards the peridotites, as predicted by thermodynamic constraints and experimental works (Lambart et al., 2012 and refs.). However, the extent to which such melts (which are not necessarily enriched in LREE, by the way) may lead to the development of kinetic fractionation similar to those observed by the authors is far from being known and is not mentioned by Stracke and Bourdon (2009), who assume that “melting of lithologically heterogeneous sources and melt extraction without reaction between the pyroxenite melt and the surrounding peridotite” for the purpose of their modelling. So, I suggest to remove or modify the sentence at lines 304-309. It would be certainly interesting to apply the simulation of kinetic fractionation to partial melts derived from recycled eclogites or pyroxenites percolating into peridotites but this is obviously beyond the scope of this paper. Actually, the ability of pyroxenite-derived melts to migrate through the mantle without reacting (or without being totally consumed by the reaction) is debated and depends on several factors besides the silica content, e.g. the length scale of the heterogeneity and a regime of melt transport allowing for large-scale reactive porous flow, as discussed in detail by Stracke and Bourdon (2009).

OTHER MINOR REMARKS

Line 48: “crust” or “lithosphere” is missing after oceanic

Line 97: dunites are not considered a common constituent of the melting region; instead, they may develop as replacive rocks/conduits in response to melt-rock reaction involving ascending melts

Line 229: “at greater depths than” instead of “at greater depths that”?

Line 297: radiogenic Sr isotope compositions are not a prerequisite of crust-derived heterogeneities if they are derived from MORB-type crust, whereas the general old age of recycling of such lithologies requires the development of radiogenic Pb signatures.

Reviewer #3 (Remarks to the Author):

Summary

The authors present a new rare-earth-element dataset of clinopyroxene of mantle xenoliths which were brought to the surface by Cenozoic alkali basalts of the Oran volcanic field in the Tell Atlas, NW Algeria. Extreme crystal-scale heterogeneity and Eu anomalies are observed in these mantle peridotite clinopyroxene (cpx). Diffusion modelling of REEs are addressed by the authors to argue that diffusional fractionation between clinopyroxene and melts can account for such intra-crystal heterogeneity and generate Eu anomalies. Therefore, the authors claim that positive Eu and Sr anomalies found in MORBs and OIBs can be generated by such an elemental diffusion process and challenge the “ghost plagioclase” (recycled oceanic lower crust) model proposed by Sobolev et al. (2000) in terms of Sr-rich inclusions.

The idea is new and the arguments are generally easy to follow. The diffusion model is suitable for the elemental heterogeneity and Eu anomalies for these xenolith cpx. However, this doesn't mean that their arguments can convince the mantle geochemistry community that the “ghost plagioclase” can be generated by diffusion process. Some important issues, e.g. discussions of Sr anomaly and other possibilities, are obviously ignored in the manuscript and there is no way to evaluate their modelling (crucial details of the modelling process are not provided), so I suggest this paper is not suitable for publication in Nature Communications.

Major Points:

1. To argue against the “ghost plagioclase” model, not only Eu anomaly, but also Sr anomaly should be generated by diffusion modelling, because positive Sr anomaly is much more pronounced than positive Eu anomaly in the melt inclusions. Unfortunately, the manuscript hasn't provided the modelling for Sr anomaly.
2. For melt inclusions with signatures of “ghost plagioclase”, they are severely depleted of most incompatible elements (Ba, Th, Nb, K, La, Ce, Nd and Zr), the diffusion model of this manuscript hasn't provided any explanation for such a depleted signature.
3. The recycled (oceanic lower crust) model is not the only mechanism proposed to generate the elemental signatures of “ghost plagioclase”. In recent years, there are increasing evidence that melt-rock interactions in (or contamination by) the oceanic lower crust can generate Sr and Eu anomalies of mantle melts (e.g. Peterson et al., 2014; Borisova et al., 2017; Peterson et al., 2019; Anderson et al., 2021), which should be discussed and carefully evaluated in this paper.

Reponse to reviewer #1

This paper by Tilhac and coauthors documents mineral scale evidence for chemical disequilibrium in the mantle sources of MORB and OIB. In particular, the presence of positive Eu anomalies only in clinopyroxene domains with mildly elevated LREE (Fig. 1a) and medium Eu concentrations (Fig. 2) is compelling. The numerical modeling done by the authors also reproduces the observations on a first order. The authors used their findings to argue against the suggestion of recycled plagioclase-rich lower crust (ghost plagioclase) in the mantle. Overall, I agree that positive Eu or Sr anomalies, in the absence of other evidence, cannot be taken as definitive evidence for recycled plagioclase. This is something we also suggested a few years ago, but we did not have the sort of intriguing evidence reported here. The paper is well written. I have a few concerns that need to be addressed.

First of all, how do we exclude the possibility that the disequilibrium Eu signatures were produced by the interactions between the xenoliths and the recent alkali basaltic magmas that carried the xenoliths to the surface? This is important. If the disequilibrium signatures were produced by these recent melt-mineral interactions, it would compromise the connection between the authors' observations and the disequilibrium mantle processes relevant to MORB and OIB genesis.

- It is somewhat unclear whether the reviewer refers to pre-, syn-eruption or emplacement processes (the word *magma* for instance hints at shallow processes). In any case, the studied xenoliths exhibit mineral assemblages and textures excluding that the disequilibrium Eu signatures were produced by recent interaction with their host basalts. In particular, shallow interactions can be clearly ruled out because their consequences are well known to produce characteristic features (*e.g.* Yaxley and Kamenetsky, 1999; Shea *et al.*, 2022) such as melt pockets and metasomatic assemblages (*e.g.* amphibole, phlogopite, apatite) not observed here (see **lines 96-97** and **Appendix A1 – Sample details**). Instead, the studied xenoliths record high-temperature conditions (*i.e.* up to up 1165°C) comparable to the mantle source of basaltic magmas. Our observations thus provide the first natural evidence that diffusive fractionation is effective under such conditions and that it can produce positive Eu anomalies within time constraints compatible with basalt genesis.

Second, in the measured clinopyroxene domains, strongly positive Eu anomalies coexist with moderate LREE enrichments (Fig. 1a) whereas in the modeling results strongly positive Eu anomalies are always associated with moderate LREE depletions (Fig. 1b). Why? Is it because the melt component assumed in the modeling is not sufficiently enriched in LREE? Or the timescales chosen (20 kyr and 5 kyr) did not capture the transient signatures?

- This question is based on an observation that was not described in full detail in the previous version of the manuscript due to length constraints. Reproducing the combination in our samples of moderated LREE enrichment and positive Eu anomalies requires fast diffusive re-equilibration for chromatographic fractionation of the LREE (*i.e.* fast diffusivities, small grain size, low porosity) while preserving sufficient chemical gradient (*i.e.* slow diffusivities, etc.) for the kinetic fractionation of Eu. Instead of adjusting the REE diffusivities and percolation parameters to fully match the sample data, we preferred to use conservative model settings that are more generally extrapolatable to the source of oceanic basalts (see **Appendix B2 – Model parameters and inputs**). The validity of this assumption is supported by the fact that MORB and OIB melt inclusions displaying ghost plagioclase signatures are characterized by LREE depletion, which is also shown by our models. This being said, our modelling results for 5 kyr do capture the particular feature pointed out by the reviewer (see the dashed lines in **Fig. 1b**), as now detailed in **lines 178-182**.

Lines 52-53. I'm not sure Niu and O'Hara (2009) meant to suggest that the missing Eu was retained in the DMM during the extraction of the continental crust. They probably also interpreted it as evidence for some sort of crustal recycling instead of a single-stage process.

- The implications of Niu and O'Hara (2009) are not straightforward. They stated "*These new observations, which support the notion that the DMM and BCC are complementary in terms of the overall abundances of incompatible elements, offer new insights into the crust–mantle differentiation. These observations are best explained by partial melting of amphibolite of MORB protolith during continental collision, which produces andesitic melts with a remarkable compositional [...] similarity to the BCC.*" Some (oceanic) crustal recycling is indeed invoked, but the complementarity of the DMM and CC implies that the missing Eu of the CC is retained in the DMM. We have clarified the corresponding statement in our introduction accordingly (**lines 52-54**).

Fig. 3b. Would be more intuitive to plot bulk Eu anomaly as a function of time instead of the opposite?

- We have modified **Fig. 3b** based on the reviewer's suggestion.

Reponse to reviewer # 2

The presence of Eu positive anomalies, frequently coupled to prominent Sr (and Ba) enrichments, in primitive oceanic basalts, glasses and melt inclusions from both MORB and OIB settings has been commonly attributed to the former presence of plagioclase in gabbroic cumulates from the oceanic crust recycled back into in the mantle (the so called "ghost plagioclase signature"). Other explanations claimed interaction with plagioclase-bearing cumulate sequences of the modern oceanic lithosphere or took into account disequilibrium processes during partial melting. The numerical model proposed in this paper, applied to a natural occurrence of peridotite xenoliths from the Oran volcanic field (Algeria), provides a brilliant alternative solution to generate such Eu anomalies, when the crustal recycling involvement is not adequately supported by other concomitant evidences (e.g. Pb or O isotopes).

In particular, the most relevant result achieved in this paper demonstrates that the Eu enrichment in clinopyroxenes from potential mantle sources of oceanic basalts may be reproduced by a diffusive fractionation process. The authors adopt the Oran xenoliths as a proxy of MORB-OIB mantle sources characterized by the preservation of transient positive Eu anomaly in clinopyroxene and reproduce through numerical simulation the striking intrasample variability of their REE patterns (or even zoning of single crystals).

In my opinion, this is an elegant, well written high quality paper based on a solid and sophisticated trace element modelling. I do not have particular remarks about manuscript organization and clarity of the abstract. I agree with the authors that the Eu anomalies in oceanic basalts may have different origins, not necessarily related to a role of plagioclase. The meaning of Eu enrichments in primitive melts (especially from MORB settings) may be reconsidered in the light of the new model presented here. Overall, the consistency of the adopted model, the originality of the approach and the accuracy of data treatment are remarkable and this work may certainly stimulate further debate on the paradigm of crustal recycling relying on trace element signatures like Eu and Sr.

The most relevant previous works have been adequately considered with few exceptions that I will underline below. To my knowledge, the conclusions of this manuscript are absolutely novel and original.

I am basically favourable to the publication of this manuscript provided that the authors address some issues reported in the following:

- Although no such systematic and detailed description of the Eu anomalies nor modelling have been attempted before, the occurrence of such anomalies in clinopyroxenes from spinel orogenic peridotites have been already reported elsewhere (e.g. see Rampone et al., 1995, *J. Petrology*, DOI: 10.1093/ptrology/36.1.81; Nain et al., 2018, <https://doi.org/10.1016/j.lithos.2018.04.001>; Marchesi et al. 2013, <http://dx.doi.org/10.1016/j.epsl.2012.11.047>), so I suggest to add a brief report in this regard. In the Ronda lherzolites, the geochemical signature of ghost plagioclase in the form of positive Eu and Sr anomalies in whole-rock and clinopyroxene was attributed to the transfer of a low-pressure crustal imprint from the adjacent recycled pyroxenite to hybridized peridotite, whereas in the other cases no straightforward explanations were provided.

- As suggested by the reviewer, we have added a reference (**lines 133-137**) to Marchesi *et al.* (2013 and references therein) which reports the occurrence of Eu anomalies in clinopyroxenes from spinel orogenic peridotites that is the most relevant to this work and includes the other references mentioned by the reviewers.

- The analytical methodology is sound and the quality of data is very good. However, I suggest to add in Appendix A a table including the REE composition of the clinopyroxenes (average or representative values) used for the calculations. This would help to make the results reproducible. Also, a brief description of the samples taken from Hidas et al., 2019 and employed for the modelling could be useful and more informative for the reader.

- Detailed sample descriptions (**Appendix A1 – Sample details**) and all the REE compositions in clinopyroxene as well as standard analyses (**Appendix A – Table 1**) have accordingly been added to the revised submission.

- The procedures adopted for the calculations and data treatment is exhaustively explained in the Appendix B but the results are not easily reproducible. Several references are cited (Oliveira et al., 2016, 2018, 2020; Tihac et al., 2020) but no clear reference to a downloadable MATLAB code is reported. In alternative, a sentence like “A MATLAB (?) implementation of the described procedure is available from the authors upon request” could be added.

- For full reproducibility of the results, all model inputs and parameters are now listed in **Appendix B2 – Table 2** in addition to the exhaustive description of the modelling approach and formulation in **Appendix B1**. As suggested by the reviewer, the MATLAB code has now been made available from the GitHub repository of the corresponding author (<https://github.com/romaintilhac/percolation-diffusion.git>).

- The Eu positive anomalies of the melts are generally coupled to Sr (and Ba) enrichments (Hofmann and Jochum, 1996; Chauvel et al., 2000; Kokfelt et al., 2006; Peterson et al., 2014; Tang et al., 2017; Anderson et al., 2021). However, Sr abundances were not taken into account for modelling of diffusive fractionation. It would be interesting to see how Sr (which has been certainly measured in the cpx from the xenoliths) behaves during the kinetic fractionation affecting the REE.

- As detailed in our introduction (e.g. **lines 40-49**), ghost plagioclase signatures in primitive basalts have indeed been traditionally interpreted from positive anomalies of various divalent cations (*i.e.* Eu, Sr, Ba). Among these elements, Sr and Ba are highly incompatible compared to Eu. In highly aggregated magmas such as basalts, this characteristic can be overlooked. However, in residual mantle peridotites, highly incompatible elements are overprinted by late processes (*i.e.* formation of fluid/melt inclusions, melt trapping along grain boundaries, etc.) that obscure the melting history originally recorded (Bedini and Bodinier, 1999; Garrido *et al.*, 2000; Bodinier and Godard, 2014). This is a long known, but still relevant issue (see the recent *Nature Correspondence* by Alard *et al.*, 2022), and the reason why the melting record of abyssal peridotites is exclusively studied *via* REE or more compatible elements (e.g. Johnson *et al.*, 1990; Niu *et al.*, 1997; Warren and Shimizu, 2010; Brunelli *et al.*, 2014;

Liang and Liu, 2016; Warren, 2016), whereas incompatible elements such as Sr and Ba record secondary processes.

Accordingly, focusing on REE allow us to demonstrate that the positive anomalies observed can only be produced by diffusive fractionation of Eu in our peridotite samples, which is highlighted as a compelling piece of evidence by reviewers #1 and #2. Therefore, only Eu anomalies can unambiguously identify ghost plagioclase signatures produced by diffusive fractionation during melting processes.

As for the modelling, for consistency we focused on the diffusive fractionation of Eu but simulating Sr anomalies is implicit because we already use Sr as a proxy for the diffusivity of Eu^{2+} (which is currently not experimentally constrained, as explained in **lines 159-162**). Considering that our mixture of Eu^{2+} - Eu^{3+} with 10–30% Eu^{2+} produces strong positive anomalies, a model directly using Sr would produce even stronger anomalies (see **Appendix B2 – Model parameters and inputs**). We find unfitting to show such results and potentially confusing for the reader as they would not unambiguously prove or disprove the ability of diffusive fractionation to generate ghost plagioclase signatures. Besides, modelling of the diffusive fractionation of Sr has already been published by Tang *et al.* (2017).

- It looks like that the occurrence of peridotite sources with (transient) Eu enrichments in their cpx may explain the generation of melts with this signature. Since the size of the Eu anomaly seems correlated with LREE enrichment of the cpx, I would expect that the derivative Eu-enriched melts are not LREE-depleted like common N-MORB. If we consider a large amount of literature data, is there a correlation between Eu/Eu^* and LREE fractionation (e.g. LaN/SmN)?

- Large literature datasets such as the one we used in **Fig. 5a** show no significant correlation between Eu/Eu^* and LREE fractionation, which reflects the fact that the contribution of peridotite sources enriched in Eu competes with the direct development of Eu anomalies by disequilibrium melting.

- The authors convincingly explain that in the MORB (and OIB) sources timescale of melt generation and redox conditions are appropriate for their model and that the LREE-enriched melts required for the diffusive fractionation could form (in the case of MORB) in a hydrous melting region deeper than the dry melting region. However, the authors state (lines 240-241) that “significant diffusive fractionation probably occurs in the low-degree, hydrous melting region that lies beneath dry peridotite solidus (Asimow and Langmuir, 2003)”. Afterwards, the rocks affected by these “transient” Eu anomalies should be involved in the dry decompression melting (and, consequently, the LREE enrichment are lost). However, is this scenario is the modelling realistic? The diffusive fractionation model (Appendix B) assumes no concomitant melting, which seem in contrast with the sentence above. Otherwise, an open system melting (e.g. Brunelli *et al.*, 2014, GCA) including the effects of kinetic fractionation could be more appropriate?

- In the two-stage scenario proposed, melt-peridotite interaction in the deep, hydrous melting regime first enriches the source in Eu (this is what we modelled) and this source may then indeed experience dry melting producing melt with positive Eu anomalies (this is what Tang *et al.*, 2017 modelled). So, no concomitant melting is used (see **Appendix B1 – General approach and model formulation**) because our percolation-diffusion model focuses on the first stage (*i.e.* melt-peridotite interaction). As for the second stage, it is already known from Tang *et al.* (2017) models and intuitive from the disequilibrium melting formulation of Qin (1992) that it can only either produce or further enhance the Eu anomalies in the melt. We have further clarified these aspects in the last discussion (in particular, **lines 274-280**).

Regarding the use of an open-system melting model such as that of Brunelli *et al.* (2014), it is not suitable for kinetic fractionation. It is indeed based on the OSM formulation of Ozawa (2001) that is intrinsically designed for a steady state and not able to reproduce the kinetic effect of transient features.

- From Appendix B: to the purpose of the calculations “ ..it is assumed that no chemical reaction occurs between peridotite and melt”. If I understand properly, the cpx (or some cpx) from the xenoliths derive from opx replacement by reactive melt percolation, as supported by EBSD data (Hidas *et al.*, 2019). Does this affect somehow the model?

- The effects of modal changes during percolation are already well known and predictable (*e.g.* Godard *et al.*, 1995). In particular, increasing the cpx/opx ratio during percolation would only result in further impeding the chromatographic effect (as cpx dominates the bulk REE partition coefficients). We thus assumed that no phase/modal change occurs during the percolation-diffusion process in order to highlight the kinetic effect of diffusional re-equilibration on the REE compositions only without involving any secondary parameters. This clarification has been added to **Appendix B1 – General approach and model formulation** in the revised submission.

- Lines 301-309: melting of silica-excess recycled mafic lithologies produces relatively SiO₂-rich melts which are highly reactive towards the peridotites, as predicted by thermodynamic constraints and experimental works (Lambart *et al.*, 2012 and refs.). However, the extent to which such melts (which are not necessarily enriched in LREE, by the way) may lead to the development of kinetic fractionation similar to those observed by the authors is far from being known and is not mentioned by Stracke and Bourdon (2009), who assume that “melting of lithologically heterogeneous sources and melt extraction without reaction between the pyroxenite melt and the surrounding peridotite” for the purpose of their modelling. So, I suggest to remove or modify the sentence at lines 304-309. It would be certainly interesting to apply the simulation of kinetic fractionation to partial melts derived from recycled eclogites or pyroxenites percolating into peridotites but this is obviously beyond the scope of this paper. Actually, the ability of pyroxenite-derived melts to migrate through the mantle without reacting (or without being totally consumed by the reaction) is debated and depends on several factors besides the silica content, *e.g.* the length scale of the heterogeneity and a regime of melt transport allowing for large-scale reactive porous flow, as discussed in detail by Stracke and Bourdon (2009).

- The sentences mentioned here are indeed reaching towards less direct implications of this work but we believe that they are worth mentioning. The reviewer’s point regarding the extractability/ reactivity of pyroxenite-derived melts is very fair, but we do not imply that these melts can be extracted and ascent over significant distances. It is the sheer fact that they are likely to react with peridotites that makes them preferential ingredients to promote kinetic fractionation in peridotite minerals (and therefore relevant to this work). We have clarified this point in the corresponding statement (**lines 309-314**).

- Line 48: “crust” or “lithosphere” is missing after oceanic

- Corrected.

- Line 97: dunites are not considered a common constituent of the melting region; instead, they may develop as replacive rocks/conduits in response to melt-rock reaction involving ascending melts

- Indeed. We have modified this statement accordingly (now **lines 98-99**).

- Line 229: “at greater depths than” instead of “at greater depths that”?

- Corrected (now **line 232**).

- Line 297: radiogenic Sr isotope compositions are not a prerequisite of crust-derived heterogeneities if they are derived from MORB-type crust, whereas the general old age of recycling of such lithologies requires the development of radiogenic Pb signatures.

- We have clarified the statement accordingly (now **line 300-302**).

Response to reviewer #3

The authors present a new rare-earth-element dataset of clinopyroxene of mantle xenoliths which were brought to the surface by Cenozoic alkali basalts of the Oran volcanic field in the Tell Atlas, NW Algeria. Extreme crystal-scale heterogeneity and Eu anomalies are observed in these mantle peridotite clinopyroxene (cpx). Diffusion modelling of REEs are addressed by the authors to argue that diffusional fractionation between clinopyroxene and melts can account for such intra-crystal heterogeneity and generate Eu anomalies. Therefore, the authors claim that positive Eu and Sr anomalies found in MORBs and OIBs can be generated by such an elemental diffusion process and challenge the “ghost plagioclase” (recycled oceanic lower crust) model proposed by Sobolev et al. (2000) in terms of Sr-rich inclusions.

The idea is new and the arguments are generally easy to follow. The diffusion model is suitable for the elemental heterogeneity and Eu anomalies for these xenolith cpx. However, this doesn't mean that their arguments can convince the mantle geochemistry community that the “ghost plagioclase” can be generated by diffusion process. Some important issues, e.g. discussions of Sr anomaly and other possibilities, are obviously ignored in the manuscript and there is no way to evaluate their modelling (crucial details of the modelling process are not provided), so I suggest this paper is not suitable for publication in Nature Communications.

- As a general reply to this criticism, we would first like to highlight the enthusiasm of the other reviewers who find our observations “*compelling*” and “*the consistency of the adopted model, the originality of the approach and the accuracy of data treatment [...] remarkable*”. They also recognize in our interpretations a “*brilliant alternative solution to generate such Eu anomalies*”. This contradicts reviewer #3's doubts about the ability of our arguments to “*convince the mantle geochemistry community*”.

We also strongly disagree that we might have ignored alternative possibilities. As noted by reviewer #2 (“*most relevant previous works have been adequately considered*”), our introduction provides a fair summary of all the different interpretations of the ghost plagioclase signature and related proxies in oceanic basalts (e.g. **lines 40-54**). Providing such a balanced review is an important prerequisite for the reader to be able to clearly identify our key contribution to this debate. We do not pretend to disprove any hypothesis but simply lend support to an alternative that had only been theoretically envisaged before, by demonstrating that Eu anomalies can be produced by kinetic fractionation in natural peridotites under conditions compatible with basalt genesis. In contrast, we find the comments of this reviewer partial and favouring the crustal recycling model against other alternatives, which is not a fair reflection of the diversity of opinions in the community.

As for the modelling details, they are now all fully provided in **Appendix B** and the MATLAB code is available.

1. To argue against the “ghost plagioclase” model, not only Eu anomaly, but also Sr anomaly should be generated by diffusion modelling, because positive Sr anomaly is much more pronounced than positive Eu anomaly in the melt inclusions. Unfortunately, the manuscript hasn't provided the modelling for Sr anomaly.

- Modelling Sr anomalies would be trivial since marked Eu anomalies are already obtained using only 10–30% Eu²⁺ with Sr as a proxy (see lines **lines 159-162**). Sr anomalies in melt inclusions are indeed more pronounced than Eu, but diffusive fractionation can easily produce Sr anomalies of the same order of magnitude, and even greater (as previously modelled by Tang *et al.*, 2017), even when using conservative estimates for diffusivities (see for instance their Fig. 5). Besides, as explained in detail in a previous comment above, primary features related to melting are only likely to be preserved by compatible or moderately incompatible elements such as REE (among which Eu), and not by Sr or more incompatible elements. Again, the vast majority of papers investigating the melting regime of oceanic basalts through mantle peridotites also focused on REE (*e.g.* Warren, 2016 and references therein).

2. For melt inclusions with signatures of “ghost plagioclase”, they are severely depleted of most incompatible elements (Ba, Th, Nb, K, La, Ce, Nd and Zr), the diffusion model of this manuscript hasn’t provided any explanation for such a depleted signature.

- As far as REE such as La, Ce and Nd are concerned, our modelling results are consistent with this observation: the most pronounced Eu anomalies are found in the most depleted cases (see **Fig. 1b**). Yet, indeed, we do not pretend to explain all the geochemical features of melt inclusions with ghost plagioclase signatures such as their depletion in highly incompatible fluid-mobile elements and some of the HFSE, nor their isotopic characteristics. These are beyond the scope of this paper, which is *only* a paper reporting the first natural evidence of diffusive fractionation of Eu among the REE and exploring its bearing on the interpretation of the ghost plagioclase signature.

3. The recycled (oceanic lower crust) model is not the only mechanism proposed to generate the elemental signatures of “ghost plagioclase”. In recent years, there are increasing evidence that melt-rock interactions in (or contamination by) the oceanic lower crust can generate Sr and Eu anomalies of mantle melts (*e.g.* Peterson *et al.*, 2014; Borisova *et al.*, 2017; Peterson *et al.*, 2019; Anderson *et al.*, 2021), which should be discussed and carefully evaluated in this paper.

- All of these alternatives are already presented in the introduction (*e.g.* **lines 46-49**) and the references mentioned by the reviewer are all cited, except Borisova *et al.* (2017) which does not deal with Sr or Eu anomalies. The role of oceanic crust contamination is also specifically considered in the last discussion (**lines 304-306**) where we emphasize that most of the interpretations of the ghost plagioclase signature are not mutually exclusive.

Our aim is merely to propose another perspective, perhaps underrepresented in the community, which consists in taking into consideration kinetic effects, and particularly so when these are unambiguously supported by natural observations such as the one reported here. This is certainly one of the reasons why this contribution will stimulate further debate on the interpretation of trace-element proxies.

REFERENCES

- Alard, O., Halimulati, A., and Demouchy, S., 2022, Look between the grains: *Nature Geoscience*, v. 15, no. 11, p. 856-857.
- Bedini, R., and Bodinier, J.-L., 1999, Distribution of incompatible trace elements between the constituents of spinel peridotite xenoliths: ICP-MS data from the East African Rift: *Geochimica et Cosmochimica Acta*, v. 63, no. 22, p. 3883-3900.
- Bodinier, J.-L., and Godard, M., 2014, Orogenic, Ophiolitic, and Abyssal Peridotites, *in* Turekian, K. K., ed., *Treatise on Geochemistry (Second Edition)*: Oxford, Elsevier, p. 103-167.

- Borisova, A. Y., Bohron, W. A., and Grégoire, M., 2017, Origin of primitive ocean island basalts by crustal gabbro assimilation and multiple recharge of plume-derived melts: *Geochemistry, Geophysics, Geosystems*, v. 18, no. 7, p. 2701-2716.
- Brunelli, D., Paganelli, E., and Seyler, M., 2014, Percolation of enriched melts during incremental open-system melting in the spinel field: A REE approach to abyssal peridotites from the Southwest Indian Ridge: *Geochimica et Cosmochimica Acta*, v. 127, p. 190-203.
- Garrido, C. J., Bodinier, J.-L., and Alard, O., 2000, Incompatible trace element partitioning and residence in anhydrous spinel peridotites and websterites from the Ronda orogenic peridotite: *Earth and Planetary Science Letters*, v. 181, no. 3, p. 341-358.
- Godard, M., Bodinier, J.-L., and Vasseur, G., 1995, Effects of mineralogical reactions on trace element redistributions in mantle rocks during percolation processes: A chromatographic approach: *Earth and Planetary Science Letters*, v. 133, no. 3-4, p. 449-461.
- Johnson, K., Dick, H. J., and Shimizu, N., 1990, Melting in the oceanic upper mantle: an ion microprobe study of diopsides in abyssal peridotites: *Journal of Geophysical Research: Solid Earth (1978-2012)*, v. 95, no. B3, p. 2661-2678.
- Liang, Y., and Liu, B., 2016, Simple models for disequilibrium fractional melting and batch melting with application to REE fractionation in abyssal peridotites: *Geochimica et Cosmochimica Acta*, v. 173, p. 181-197.
- Marchesi, C., Garrido, C. J., Bosch, D., Bodinier, J.-L., Gervilla, F., and Hidas, K., 2013, Mantle refertilization by melts of crustal-derived garnet pyroxenite: Evidence from the Ronda peridotite massif, southern Spain: *Earth and Planetary Science Letters*, v. 362, p. 66-75.
- Niu, Y., Langmuir, C. H., and Kinzler, R. J., 1997, The origin of abyssal peridotites: a new perspective: *Earth and Planetary Science Letters*, v. 152, no. 1, p. 251-265.
- Niu, Y., and O'Hara, M. J., 2009, MORB mantle hosts the missing Eu (Sr, Nb, Ta and Ti) in the continental crust: New perspectives on crustal growth, crust-mantle differentiation and chemical structure of oceanic upper mantle: *Lithos*, v. 112, no. 1-2, p. 1-17.
- Qin, Z., 1992, Disequilibrium partial melting model and its implications for trace element fractionations during mantle melting: *Earth and Planetary Science Letters*, v. 112, no. 1, p. 75-90.
- Shea, J. J., Ezad, I. S., Foley, S. F., and Lanati, A. W., 2022, The Eastern Australian Volcanic Province, its primitive melts, constraints on melt sources and the influence of mantle metasomatism: *Earth-Science Reviews*, v. 233.
- Tang, M., McDonough, W. F., and Ash, R. D., 2017, Europium and strontium anomalies in the MORB source mantle: *Geochimica et Cosmochimica Acta*, v. 197, p. 132-141.
- Warren, J. M., 2016, Global variations in abyssal peridotite compositions: *Lithos*, v. 248-251, p. 193-219.
- Warren, J. M., and Shimizu, N., 2010, Cryptic Variations in Abyssal Peridotite Compositions: Evidence for Shallow-level Melt Infiltration in the Oceanic Lithosphere: *Journal of Petrology*, v. 51, no. 1-2, p. 395-423.
- Yaxley, G. M., and Kamenetsky, V., 1999, In situ origin for glass in mantle xenoliths from southeastern Australia: insights from trace element compositions of glasses and metasomatic phases: *Earth and Planetary Science Letters*, v. 172, no. 1-2, p. 97-109.

REVIEWERS' COMMENTS

Reviewer #1 (Remarks to the Author):

I'm satisfied with the revisions made by the authors and have no further comments.

Ming Tang

12/9

Reviewer #3 (Remarks to the Author):

Now the authors have successfully convinced me that diffusion-induced fractionation of REEs in the upper mantle, which can be an alternative origin of the positive Eu anomaly. I agree with their claim that 'most of the interpretations of the ghost plagioclase signature are not mutually exclusive.' In the revised version, they have provided the REE data of these xenolith clinopyroxenes, the modelling details and the MATLAB code. I'm satisfied with their revision and their answers/interpretations. My only suggestion is that not only REEs but also other trace elements of these clinopyroxenes should be published in this paper. These data seem useless in this paper, but might be useful for the readers.

Some typos:

Line 94: add 'mantle' before 'xenoliths'.

Line 207: delete '(b)' in this sentence.

Reponse to reviewer # 1

I'm satisfied with the revisions made by the authors and have no further comments.

Reponse to reviewer # 2

Now the authors have successfully convinced me that diffusion-induced fractionation of REEs in the upper mantle, which can be an alternative origin of the positive Eu anomaly. I agree with their claim that 'most of the interpretations of the ghost plagioclase signature are not mutually exclusive.' In the revised version, they have provided the REE data of these xenolith clinopyroxenes, the modelling details and the MATLAB code. I'm satisfied with their revision and their answers/interpretations. My only suggestion is that not only REEs but also other trace elements of these clinopyroxenes should be published in this paper. These data seem useless in this paper, but might be useful for the readers.

As explained in our previous response, we here exclusively report analytical and modelled REE data for the sake of consistency. Providing an exhaustive geochemical characterization of these clinopyroxenes that includes more trace elements is beyond the scope of this work.

Some typos:

Line 94: add 'mantle' before 'xenoliths'.

Done.

Line 207: delete '(b)' in this sentence.

This figure caption has been modified accordingly.